# Changes in hospitalizations and emergency department respiratory viral diagnosis trends before and during the COVID-19 pandemic in Ontario, Canada

**Steven Habbous**[1,2]*, **Susy Hota**[3], **Vanessa G. Allen**[1,3,4], **Michele Henry**[1], **Erik Hellsten**[1]

**1** Ontario Health (Strategic Analytics), Toronto, Ontario, Canada, **2** Epidemiology & Biostatistics, Western University, London, Ontario, Canada, **3** Department of Medicine, Division of Infectious Diseases, University Health Network, Toronto, Ontario, Canada, **4** Department of Microbiology, Sinai Health/ University Health Network, Toronto, Ontario, Canada

* steven.habbous@ontariohealth.ca

## Abstract

### Introduction

Population-level surveillance systems have demonstrated reduced transmission of non-SARS-CoV-2 respiratory viruses during the COVID-19 pandemic. In this study, we examined whether this reduction translated to reduced hospital admissions and emergency department (ED) visits associated with influenza, respiratory syncytial virus (RSV), human metapneumovirus, human parainfluenza virus, adenovirus, rhinovirus/enterovirus, and common cold coronavirus in Ontario.

### Methods

Hospital admissions were identified from the Discharge Abstract Database and exclude elective surgical admissions and non-emergency medical admissions (January 2017-March 2022). Emergency department (ED) visits were identified from the National Ambulatory Care Reporting System. International Classification of Diseases (ICD-10) codes were used to classify hospital visits by virus type (January 2017-May 2022).

### Results

At the onset of the COVID-19 pandemic, hospitalizations for all viruses were reduced to near-trough levels. Hospitalizations and ED visits for influenza (9,127/year and 23,061/year, respectively) were nearly absent throughout the pandemic (two influenza seasons; April 2020-March 2022). Hospitalizations and ED visits for RSV (3,765/year and 736/year, respectively) were absent for the first RSV season during the pandemic, but returned for the 2021/2022 season. This resurgence of hospitalizations for RSV occurred earlier in the season than expected, was more likely among younger infants (age ≤6 months), more likely among older children (aged 6.1–24 months), and less likely to comprise of patients residing in higher areas of ethnic diversity (p<0.0001).

**Data Availability Statement:** Ontario Health is prohibited from making the data used in this research publicly accessible if it includes potentially identifiable personal health information and/or

personal information as defined in Ontario law, specifically the Personal Health Information Protection Act (PHIPA) and the Freedom of Information and Protection of Privacy Act (FIPPA). Due to these legal and ethical restrictions, data will not be made publicly available. However, upon request, data de-identified to a level suitable for public release may be provided. Requests can be made to OH-CCO_Research@ontariohealth.ca.

**Funding:** The authors received no specific funding for this work.

**Competing interests:** The authors have declared that no competing interests exist.

## Conclusion

During the COVID-19 pandemic, there was a reduced the burden of other respiratory infections on patients and hospitals. The epidemiology of respiratory viruses in the 2022/23 season remains to be seen.

## Introduction

Every year in Canada, there are seasonal surges in infections involving a variety of respiratory viruses that are driven by changes in human behaviours (e.g. time spent indoors) and environmental conditions (e.g. temperature, humidity) [1].

To aid in the effort to treat, prevent, and control spread, respiratory viruses of public health importance are tracked through surveillance systems, include influenza, respiratory syncytial virus (RSV), common cold (formerly seasonal) coronavirus [2], human metapneumovirus (hMPV), rhinovirus/enterovirus, and human parainfluenza virus (hPINV) [3, 4]. While surveillance data on incidence of these respiratory viruses is often publicly available, hospital volumes is a more direct measure of the burden of respiratory viruses on health care systems and patients, particularly as people may have avoided going to clinics for testing due fears of contracting SARS-CoV-2 [5].

Since the start of the COVID-19 pandemic, the number of people affected by influenza, RSV, and the broader array of respiratory viruses decreased in many countries [6–11]. Although this may be partially driven by viral interference caused by SARS-CoV-2 infection, reduced testing due to reagent shortages, and laboratory burden, the main drivers are likely modifications in human behavior as a result of public health measures (e.g. restrictions, lockdowns) and individual risk-reduction tactics (e.g. mask-wearing, avoiding crowded spaces, increased hand hygiene) to combat the COVID-19 pandemic [12–14]. At the time of writing, the majority of the published literature has focused on the initial pandemic year when restrictions were heightened. More recently, analysis of surveillance data from countries in the southern hemisphere suggest a resurgence in RSV as restrictions eased, but less data are available for countries in more northern climates [11].

In the present study we examined year-over-year trends in hospitalizations and emergency department (ED) visits associated with various respiratory viruses in Ontario, Canada.

## Methods

Hospital care episodes starting between January 1, 2017 and the most recent data were included (weeks starting March 25, 2022 for admissions and May 20, 2022 for emergency department (ED) visits for complete weekly counts). Ontario is Canada's most populous province (~14.5 million) and healthcare is provided under a single-payer system.

Hospital admissions were identified from the Discharge Abstract Database (DAD) and emergency department (ED) visits from the National Ambulatory Care Reporting System (NACRS). To avoid double-counting, admissions and ED visits were resolved into episodes of care after accounting for hospital transfers and readmissions as previously described [15, 16]. We modified this definition by considering planned admissions (readmit code 1) within 1 week of the previous discharge as part of the previous episode [17]. Non-emergency surgical or medical admissions were excluded.

The International Classification of Diseases, 10th revision (ICD-10) codes used to identify influenza (J09, J10.0, J10.1, J10.8, J11.0, J11.1, J11.8) and RSV (J12.1, J20.5, J21.0, B97.4) were

previously validated against the Ontario Laboratories Information System (OLIS) and used all diagnostic codes for the admission (up to 25) [18]. For influenza, the best-performing algorithm yielded 73% sensitivity and 98% specificity (FLU2) and for RSV 69% sensitivity and 99% specificity (RSV1) [18]. In the absence of validation studies for the human metapneumovirus (hMPV; J12.3, J21.1, A85.8, J20.8, B97.81), rhinovirus/enterovirus (B34.1, J20.6, B97.1), adenovirus (B34.0, B97.0), human parainfluenza virus (hPINV; J12.2, J20.4), or common cold coronavirus (B34.2, B97.2), we similarly included all diagnostic types (e.g. not restricted to the most responsible diagnosis) and extended this to ED visits as well (up to 10 diagnostic codes per record). Rhinovirus and enterovirus were grouped together since many molecular diagnostic assays do not distinguish them [3].

## Covariates

Age at admission and sex were obtained from the Registered Persons Database (RPDB). Residence in long-term care was ascribed if the patient had any primary care billing code starting with "W" from the Ontario Health Insurance Program or had a prior admission to/from a long-term care residence in the previous year. Sociodemographic characteristics were obtained using the patients' most recent postal code (RPDB) and linked to the 2016 Census (rurality) and the Ontario Marginalization Index for neighborhood-level marginalization. The Ontario Marginalization index is a linear combination of neighborhood-level socioeconomic characteristics from the Canada Census. It was estimated using a principal component analysis, generating four primary orthogonal factors categorized into quintile measures of material deprivation, residential instability, dependency, and ethnic diversity [19].

## Context

The COVID-19 pandemic was declared a global pandemic by the World Health Organization on March 11, 2020. On March 17, 2020, Ontario declared a province-wide state of emergency. At the time of analysis, data were available through the putative end of the fifth wave (dominated by the Omicron B.1.1.529 variant) of SARS-CoV-2 infections in the province. Approximate start dates of COVID-19 waves in Ontario were February 26, 2020 (wave 1 – predominately the SARS-CoV-2 wildtype variant), September 1, 2020 (wave 2 –predominately wildtype), March 1, 2021 (wave 3 –predominately the Alpha variant), August 1, 2021 (wave 4 – predominately Delta), and December 15, 2021 (wave 5 –predominately Omicron B.1.1.529) [20].

## Outcomes

During the admission episode, we examined the utilization of intensive care units (ICU) (special care unit codes 10, 30, 70, or 80) and use of respiratory support (1GZ30, 1GZ31, 1GZ32, 1GZ35, 1GZ38). Total length of stay in days was computed at the level of the entire admission episode.

## Statistics and privacy

The predicted levels of admissions or emergency department visits were constructed using a linear regression on the weekly number of visits between January 2017 and February 2020 using calendar year, month, and holiday weeks as predictors. This was then extrapolated into the COVID-19 era. By viral admission, we used logistic regression to compare patient and admission characteristics, ICU use, and respiratory support in the COVID-19 era versus the

pre-COVID-19 era, presenting odds ratios (OR) with 95% confidence intervals (CI). P-values <0.05 were considered statistically significant.

All analyses were conducted using Statistical Analysis Software (SAS Institute., Cary, NC). Values <6 were suppressed to prevent re-identification. This study was compliant with section 45(1) of PHIPA (Ontario Health is a prescribed entity): ethics review was not required.

## Results

### Influenza

Pre-pandemic (July 1, 2017 –June 30, 2019), there were 9127 admissions and 23,061 ED visits associated with influenza per year, with 2.5-times as many ED visits as admissions (Table 1).

Admissions and ED visits due to influenza had seasonal peaks in the winter months (December–March) and troughs in the summer months (July–August) (Fig 1a). At the onset of the COVID-19 pandemic, admissions due to influenza quickly dropped to trough levels and did not increase again through to the end of the study period. ED visits displayed a similar pattern with a 14-fold reduction in ED visits expected in a given influenza season (e.g. 1738 in the 2020/21 season versus 24,443 the previous year; Table 1; Fig 2a).

### Respiratory syncytial virus (RSV)

At the onset of the COVID-19 pandemic, both admissions and ED visits for RSV quickly dropped, but the timing coincided with the tail end of RSV-related seasonal hospital activity (Fig 1b). During the COVID-19 pandemic, the expected seasonal surge did not occur during the 2020/21 RSV season (21 admissions and 9 ED visits; Table 1). However, unlike influenza, RSV-related admissions returned for the 2021/22 season (September 2021 –March 2022). ED visits due to RSV in the 2021/2022 season, although not yet complete, has already surpassed previous seasonal visits (966 ED visits versus 848 in the 2018/19 RSV season), but the ratio of admissions to ED visits remained lower than pre-pandemic levels (2.3 versus 5.1), suggesting more mild cases or a reduced proclivity towards admission.

During the study period, patients admitted for RSV were mostly <24 months of age (n = 8,092; 52%) or >65 years of age (n = 4,598; 29%) (S1 Table; S1 Fig). Compared to pre-pandemic cycles (2017/18 and 2018/19 seasons), patients admitted in the 2021/22 season for RSV were more likely to comprise of infants (39% vs 30% were 0–6 months) and children (16% vs 8% were 2–5 years), but less likely to comprise patients >5 years of age (24% vs 42%) (Table 2). RSV-related admissions occurred earlier in the year: compared with January, the relative odds of admission ranged from 4.97 in July to 3.83 in December, peaking at 10.6 in October. Admissions were less likely during the months after January with available data [OR 0.29 (0.22–0.40) in February and OR 0.57 (0.42–0.78) in March]. Patients were more likely to be admitted in the 2021/22 season if they resided in a rural area [OR 1.25 (1.03–1.50)] or if they resided in a neighbourhood with a lower ethnic diversity (p<0.0001).

During the study period, ED visits were mostly prevalent among infants [n = 1932 (51%) were <6 months] and children [n = 1167 (31%) were 6–24 months] and only 147 (4%) were >65 years (S1 Table). Compared to pre-pandemic cycles (2017/18 and 2018/19 seasons), patients visiting the ED for RSV in the 2021/22 season were more likely to occur earlier in the season with a peak in October [OR 9.99 (6.47–15.5)] and was more likely to occur among children age 2–5 [OR 1.95 (1.51–2.52)] and persons aged 5–65 [OR 1.57 (1.01–2.44)] compared with infants (S2 Table). There were no other meaningful associations with sociodemographic characteristics.

**Table 1. Number of hospitalizations and emergency department visits over time, by virus type.**

| | Hospital admissions (N) | Emergency department visits (N) | Ratio (admission: ED visit) |
|---|---|---|---|
| Influenza virus | | | |
| July 2017 –June 30, 2018 | 10,711 | 25,898 | 0.41 |
| July 2018 –June 30, 2019 | 7,543 | 20,224 | 0.37 |
| July 2019 –June 30, 2020 | 6,684 | 24,443 | 0.27 |
| July 2020 –June 30, 2021 | 95 | 1783 | 0.05 |
| July 2021 –most recent | 137 | 1781 | 0.08 |
| Respiratory syncytial virus | | | |
| July 2017 –June 30, 2018 | 3340 | 624 | 5.34 |
| July 2018 –June 30, 2019 | 4190 | 848 | 4.94 |
| July 2019 –June 30, 2020 | 3722 | 718 | 5.18 |
| July 2020 –June 30, 2021 | 21 | 9 | 2.33 |
| July 2021 –most recent | 2260 | 966 | 2.34 |
| Human metapneumovirus (hMPV) | | | |
| July 2017 –June 30, 2018 | 906 | 2,364 | 0.38 |
| July 2018 –June 30, 2019 | 872 | 2,257 | 0.39 |
| July 2019 –June 30, 2020 | 649 | 1,862 | 0.35 |
| July 2020 –June 30, 2021 | 84 | 309 | 0.27 |
| July 2021 –most recent | 89 | 623 | 0.14 |
| Rhinovirus / enterovirus | | | |
| July 2017 –June 30, 2018 | 989 | 912 | 1.08 |
| July 2018 –June 30, 2019 | 1,024 | 950 | 1.08 |
| July 2019 –June 30, 2020 | 870 | 489 | 1.78 |
| July 2020 –June 30, 2021 | 142 | 124 | 1.15 |
| July 2021 –most recent | 520 | 199 | 2.61 |
| Adenovirus | | | |
| July 2017 –June 30, 2018 | 211 | 41 | 5.15 |
| July 2018 –June 30, 2019 | 195 | 52 | 3.75 |
| July 2019 –June 30, 2020 | 193 | 31 | 6.23 |
| July 2020 –June 30, 2021 | 33 | 7 | 4.71 |
| July 2021 –most recent | 93 | 34 | 2.74 |
| Human parainfluenza virus (hPINV) | | | |
| July 2017 –June 30, 2018 | 145 | <6 | n/a |
| July 2018 –June 30, 2019 | 211 | <6 | n/a |
| July 2019 –June 30, 2020 | 116 | <6 | n/a |
| July 2020 –June 30, 2021 | <6 | 0 | n/a |
| July 2021 –most recent | 58 | 27 | 2.15 |
| Common cold coronavirus | | | |
| July 2017 –June 30, 2018 | 97 | <6 | n/a |
| July 2018 –June 30, 2019 | 95 | 9 | 10.6 |
| July 2019 –June 30, 2020 | 186 | 42 | 4.43 |
| July 2020 –June 30, 2021 | 73 | 8 | 9.13 |
| July 2021 –most recent | 118 | 21 | 5.61 |

Most recent data available for analysis was March 25, 2022 for admissions and May 20, 2022 for emergency department visits for complete weekly counts

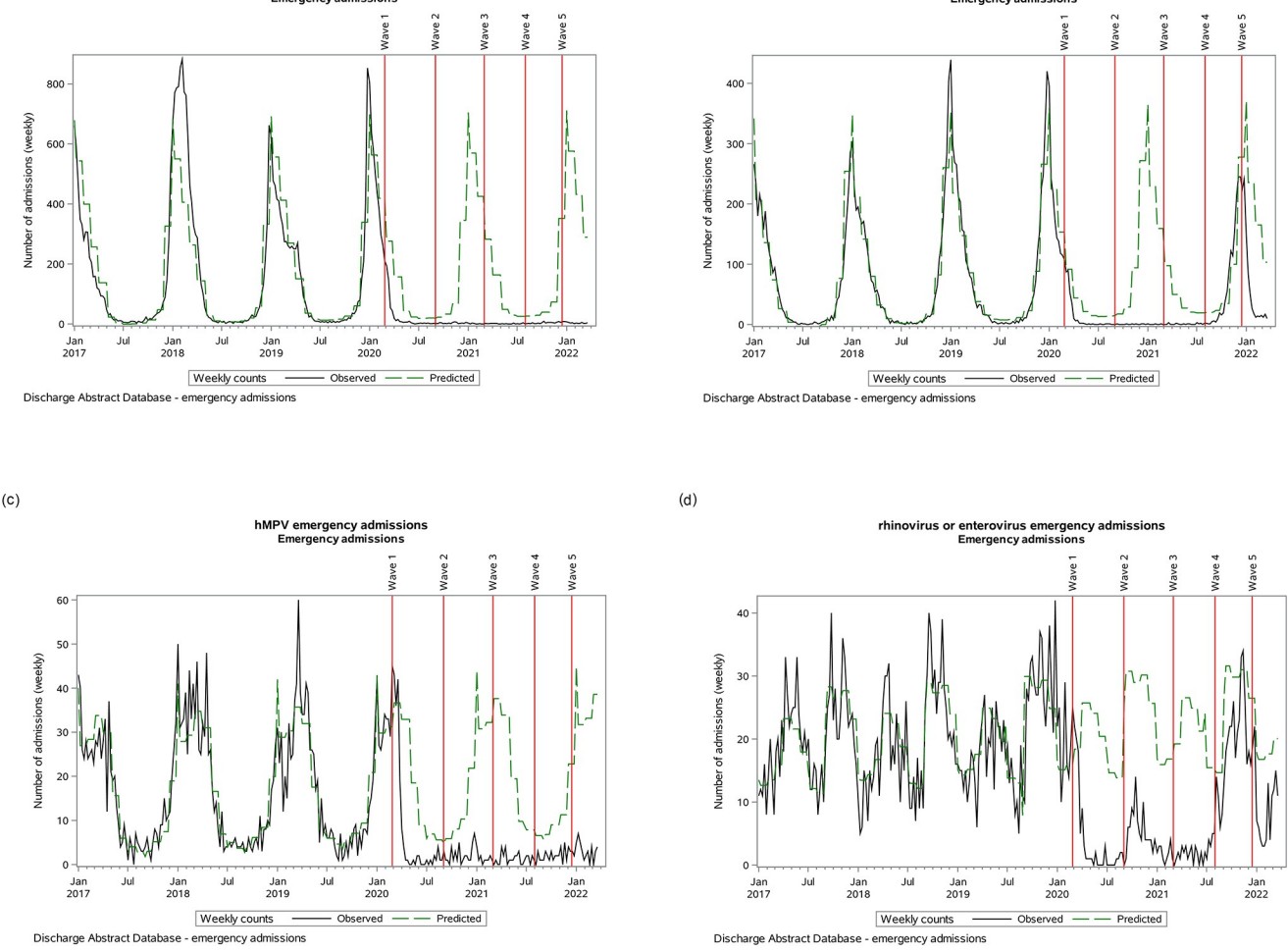

**Fig 1. Hospital admissions over time.** Hospital admissions associated with influenza (A), respiratory syncytial virus (B), human metapneumovirus (C), and rhinovirus/enterovirus.

## Human metapneumovirus (hMPV)

Admissions due to hMPV was seasonal with a winter peak extending into the spring (Fig 1c) and bimodal with respect to age (S1 Table; S1 Fig). At the onset of the COVID-19 pandemic, admissions declined quickly and remained similar to summer levels.

ED visits for hMPV displayed a peak in the early winter months and also dropped to a new trough at the start of the pandemic. Patients were a mean 42 (SD 24.0) years of age at admission [median 42.3 (IQR 23.6, 60.9) years] and 5383/8942 (60%) were female (S1 Table; S1 Fig). Despite smaller peaks occurring earlier than expected (peaks in autumn months) during the first and second hMPV seasons of the pandemic, ED visit rates remained below expected levels (Table 1).

## Rhinovirus/enterovirus

Patients admitted due to rhinovirus/enterovirus were a mean 17 (SD 27.7) years of age at admission [median 2.8 (IQR 0.9, 13.5) years] and 1793/4039 (44%) were female. Admissions peaked twice per year (Fig 1d). During the COVID-19 pandemic, admissions were below

(a)

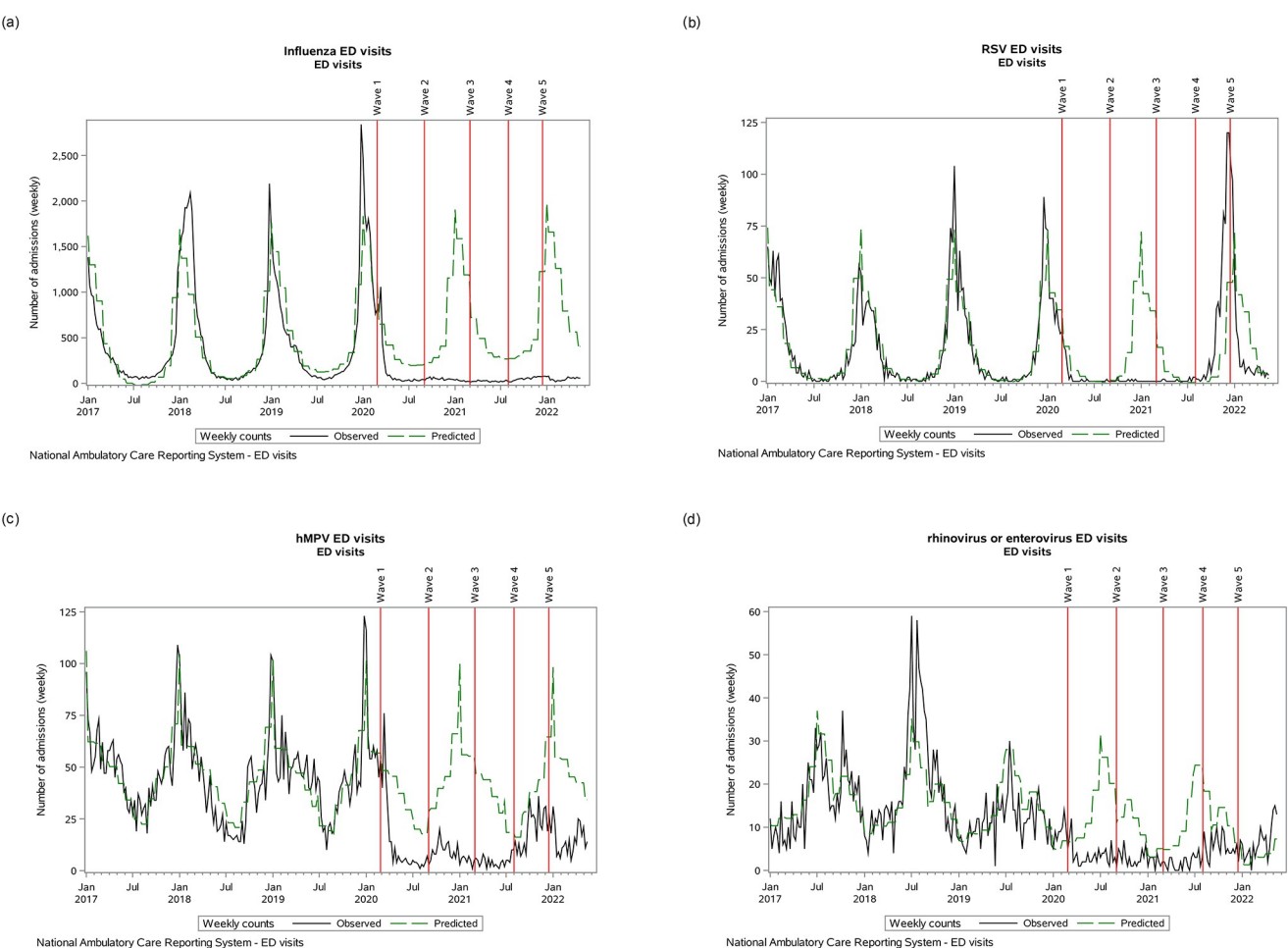

**Fig 2. Emergency department visits over time.** Emergency department visits associated with influenza (A), respiratory syncytial virus (B), human metapneumovirus (C), and rhinovirus/enterovirus.

expected volumes from previous seasons, temporarily approached baseline for the second peak of 2021, but dropped again at the onset of the fifth wave of the COVID-19 pandemic. Compared with rhinovirus/enterovirus admissions in the pre-COVID-19 era, this brief resurgence was more likely to affect children [OR 1.56 (1.13–2.14) age 6–24 months; 1.71 (1.23–2.38) age 2–5 years] and less likely to affect the elderly [OR 0.51 (0.31–0.85) age 65+] compared with infants (p<0.0001) (S3 Table). Patients hospitalized for rhinovirus/enterovirus in the COVID-19 era were less likely to reside neighbourhoods of higher dependency [OR 0.55 (0.37–0.81) for highest versus lowest quintile] and higher ethnic diversity [OR 0.43 (0.28–0.67) for highest versus lowest quintile].

Patients visiting the ED for rhinovirus/enterovirus were a mean 7.1 (SD 11.8) years of age at admission [median 2.4 (IQR 1.24, 5.8) years] and 1475/3054 (48%) were female, but ED visits remained below expectations throughout the pandemic.

## hPINV

Admissions for hPINV displayed a seasonal bimodal pattern with one peak in the winter and a second peak in the fall (S2 Fig). During the COVID-19 pandemic, the spring peak of the 2019/

**Table 2. Characteristics of admissions with respiratory syncytial virus.**

| | Pre-COVID-19 era (01/Jul/2017-30/Jun/2019) (N = 7530) | COVID-19 era (01/Jul/2021-31/Mar/2022) (N = 2262) | COVID-19 era vs. pre-COVID-19 era (N = 9592 complete case) | |
|---|---|---|---|---|
| Age | N (%) | N (%) | OR (95% CI)[a] | p-value |
| ≤6 months | 2227 (30%) | 890 (39%) | 1.0 (ref) | < .0001 |
| 6.1–24.0 months | 1424 (19%) | 454 (20%) | 0.76 (0.66–0.88) | |
| 24.1 months—5 years | 624 (8%) | 372 (16%) | 1.17 (0.99–1.38) | |
| 5.1–64.9 years | 761 (10%) | 164 (7%) | 0.74 (0.60–0.91) | |
| ≥65 years | 2444 (32%) | 382 (17%) | 0.59 (0.51–0.69) | |
| Sex | | | | |
| Female | 3826 (51%) | 1051 (46%) | 1.0 (ref) | 0.09 |
| Male | 3704 (49%) | 1211 (54%) | 1.10 (0.98–1.22) | |
| Month[b] | | | | |
| July | 11 (<1%) | 6 (<1%) | 4.97 (1.81–13.7) | |
| August | 18 (<1%) | 9 (<1%) | 4.09 (1.74–9.62) | |
| September | 39 (1%) | 40 (2%) | 8.50 (5.30–13.6) | |
| October | 115 (2%) | 166 (7%) | 10.6 (8.06–13.9) | |
| November | 556 (7%) | 580 (26%) | 7.73 (6.50–9.19) | |
| December | 1953 (26%) | 1032 (46%) | 3.83 (3.32–4.43) | |
| January | 2356 (31%) | 319 (14%) | 1.0 (ref) | < .0001 |
| February | 1321 (18%) | 56 (2%) | 0.29 (0.22–0.40) | |
| March | 688 (9%) | 54 (2%) | 0.57 (0.42–0.78) | |
| April | 327 (4%) | n/a | n/a | |
| May | 111 (1%) | n/a | n/a | |
| June | 35 (<1%) | n/a | n/a | |
| Long-term care | | | | |
| No | 7132 (95%) | 2218 (98%) | 1.0 (ref) | 0.26 |
| Yes | 398 (5%) | 44 (2%) | 0.82 (0.57–1.16) | |
| Rurality | | | | |
| Urban | 6707 (90%) | 1957 (87%) | 1.0 (ref) | 0.02 |
| Rural | 780 (10%) | 289 (13%) | 1.25 (1.03–1.50) | |
| Material deprivation | | | | |
| Lowest | 1616 (22%) | 460 (21%) | 1.0 (ref) | 0.54 |
| Mid-low | 1420 (19%) | 478 (21%) | 1.13 (0.96–1.34) | |
| Middle | 1276 (17%) | 405 (18%) | 1.07 (0.90–1.28) | |

(*Continued*)

**Table 2.** (Continued)

| Age | Pre-COVID-19 era (01/Jul/2017-30/Jun/2019) (N = 7530) | COVID-19 era (01/Jul/2021-31/Mar/2022) (N = 2262) | COVID-19 era vs. pre-COVID-19 era (N = 9592 complete case) | |
|---|---|---|---|---|
| | N (%) | N (%) | OR (95% CI)[a] | p-value |
| Mid-high | 1389 (19%) | 397 (18%) | 1.08 (0.90–1.30) | |
| Highest | 1664 (23%) | 487 (22%) | 1.16 (0.96–1.40) | |
| Residential instability | | | | |
| Lowest | 1280 (17%) | 404 (18%) | 1.0 (ref) | 0.75 |
| Mid-low | 1419 (19%) | 458 (21%) | 0.95 (0.80–1.13) | |
| Middle | 1357 (18%) | 466 (21%) | 1.03 (0.86–1.23) | |
| Mid-high | 1460 (20%) | 441 (20%) | 1.07 (0.88–1.30) | |
| Highest | 1849 (25%) | 458 (21%) | 1.02 (0.83–1.24) | |
| Dependency | | | | |
| Lowest | 1784 (24%) | 570 (26%) | 1.0 (ref) | 0.32 |
| Mid-low | 1401 (19%) | 433 (19%) | 0.94 (0.80–1.11) | |
| Middle | 1265 (17%) | 422 (19%) | 1.03 (0.86–1.23) | |
| Mid-high | 1242 (17%) | 378 (17%) | 0.89 (0.74–1.07) | |
| Highest | 1673 (23%) | 424 (19%) | 0.87 (0.72–1.05) | |
| Ethnic diversity | | | | |
| Lowest | 1175 (16%) | 427 (19%) | 1.0 (ref) | < .0001 |
| Mid-low | 1382 (19%) | 442 (20%) | 0.80 (0.67–0.96) | |
| Middle | 1494 (20%) | 466 (21%) | 0.68 (0.56–0.82) | |
| Mid-high | 1528 (21%) | 420 (19%) | 0.54 (0.44–0.67) | |
| Highest | 1786 (24%) | 472 (21%) | 0.44 (0.35–0.54) | |

[a] Odds ratio (OR) with 95% confidence interval (CI) comparing the respiratory syncytial virus resurgence (2021/2022 season) with the pre-COVID-19 seasons (2017/18 and 2018/19 seasons). OR are adjusted for calendar month, age at admission or emergency department visit, sex, rurality, deprivation quintile, instability quintile, dependency quintile, and ethnic diversity quintile

[b] Results were identical in sensitivity analysis omitting April-June from the analysis.

20 season and both peaks of the 2020/21 season were absent. The winter peak of the 2021/22 season arrived early, but the spring peak did not occur, coinciding with the onset of the fifth COVID-19 wave.

ED visits related to hPINV were generally uncommon (total 8 between January 1, 2017 and June 30, 2020. This was followed by a small but notable increase in ED visits during the autumn of the 2021/22 season (total 27 between July 2021 and March 2022).

### Adenovirus

Admissions due to adenovirus (peaks in the winter) were curtailed at the onset of the pandemic but may have partially resumed before the onset of the fifth wave of COVID-19. ED visits were generally low each year (S3 Fig).

### Common cold coronavirus

Admissions for common cold coronavirus was only slightly depressed during the first winter of the pandemic. However, at the onset of the pandemic (March 2020), the surge from the winter of 2019/2020 was prolonged. This was likely due to miscoding of the novel SARS-CoV-2 as common cold coronavirus, as the ICD-10 codes were not available for use until April 2020. This effect was more striking for ED visits (S4 Fig).

### Outcomes

Use of ICU during the admission episode was highest for hPINV (23%), followed by rhinovirus/enterovirus (18%), hMPV (14.6%), influenza (11.3%), adenovirus (9.7%), and RSV (9.2%) (Table 3). The high rate of ICU use also observed among common cold coronaviruses (18%) may be driven by misclassification with the novel SARS-CoV-2 (S4 Fig; S4 Table). Respiratory support during the admission episode was highest for admissions with common-cold coronavirus (17%) and hPINV (16%), followed by hMPV (10%), rhinovirus/enterovirus (8.9%), influenza virus (7.5%), and RSV (5.8%). Total length-of-stay was highest for hPINV (median 6.6 days), hMPV (median 4.8 days), and influenza virus (median 4.7 days). ICU use in the COVID-19 era was more likely for admissions with hMPV [OR 1.89 (1.13–3.16)]. Use of respiratory support was trending higher in the COVID-19 era for hMPV [OR 1.64 (0.91–2.94)], but decreased for RSV [OR 0.61 (0.47–0.78)] and rhinovirus/enterovirus [OR 0.53 (0.34–0.82)] (Table 3).

## Discussion

In the present study, we observed a significant reduction in hospitalizations and ED visits associated with a range of respiratory viruses during the COVID-19 pandemic. Different patterns of resurgence were observed for different viruses. Hospital visits due to influenza appear to have nearly vanished, but early signs of resurgence warrant caution [21, 22]. RSV, on the other hand, returned to baseline levels earlier than expected. Hospital visits due to hMPV, rhinovirus/enterovirus, hPINV, and adenovirus were similarly affected with either a partial or complete resurgence later in the study period.

It is reasonable to expect that the behavioural and societal modifications with non-pharmacological interventions used to combat SARS-CoV-2 infection affected the transmission of respiratory viruses differently [23, 24]. Some of this may be attributed to a differential contribution of respiratory droplet and aerosols versus fomites in transmission of various respiratory viruses. As influenza and hMPV are primarily transmitted through respiratory particles, it is no surprise that their impacts were suppressed by pandemic mask mandates and social distancing. On the other hand, hand hygiene and surface disinfection have been progressively deemphasized as SARS-CoV-2 infection prevention measures over the course of the pandemic. It is possible that viruses such as rhinoviruses and enteroviruses transmit more efficiently than other respiratory viruses, explaining their rising impacts later in the pandemic. Rhinoviruses are non-enveloped viruses, prolonging their survival on inanimate surfaces and enabling transmission by fomites and surface contact [25, 26]. Masking is also less effective against rhinoviruses than influenza [27]. Furthermore, with milder respiratory symptoms, these

**Table 3. Admission characteristics.**

| | Respiratory syncytial virus (n = 15,768) | Influenza virus (n = 29,894) | Human meta-pneumovirus (n = 3181) | Rhinovirus/enterovirus (n = 4,039) | Adenovirus (n = 829) | Common cold coronavirus (n = 722) | Human parainfluenza virus (n = 667) |
|---|---|---|---|---|---|---|---|
| Care provided[a] | | | | | | | |
| Intensive care unit[b] | 1446 (9.2%) | 3381 (11.3%) | 464 (14.6%) | 706 (17.5%) | 80 (9.7%) | 128 (17.7%) | 153 (22.9%) |
| Pre-COVID-19 era | 1199/13081 (9.2%) | - | 375/2804 (13%) | 569/3265 (17%) | 66/667 (9.9%) | 72/445 (16%) | 140/588 (24%) |
| COVID-19 era | 247/2687 (9.2%) | - | 90/377 (24%) | 137/774 (19%) | 14/162 (8.6%) | 56/277 (20%) | 13/79 (16%) |
| aOR (95% CI)[c] | 1.04 (0.88–1.24) | - | 1.89 (1.13–3.16) | 0.97 (0.75–1.27) | 0.56 (0.21–1.45) | 0.59 (0.32–1.10) | 0.43 (0.18–0.99) |
| Any respiratory support[c] | 913 (5.8%) | 2250 (7.5%) | 320 (10.1%) | 358 (8.9%) | 52 (6.3%) | 119 (16.5%) | 106 (15.9%) |
| Pre-COVID-19 era | 802/13081 (6.1%) | - | 260/2804 (9.3%) | 321/3265 (9.8%) | 43/667 (6.5%) | 76/445 (17%) | 99/588 (17%) |
| COVID-19 era | 111/2687 (4.1%) | - | 60/377 (16%) | 37/774 (4.8%) | 9/162 (5.6%) | 43/277 (16%) | 7/79 (8.9%) |
| aOR (95% CI)[c] | 0.61 (0.47–0.78) | - | 1.64 (0.91–2.94) | 0.53 (0.34–0.82) | 0.69 (0.23–2.02) | 0.31 (0.15–0.67) | 0.16 (0.04–0.68) |
| Ventilation | 860 (5.5%) | 2212 (7.4%) | 298 (9.4%) | 311 (7.7%) | 47 (5.7%) | 108 (15.0%) | 102 (15%) |
| Oxygenation | 71 (0.5%) | 45 (0.2%) | 32 (1.0%) | 80 (2.0%) | <6 | 18 (2.5%) | 8 (1.2%) |
| Total length of stay (days)[d] | | | | | | | |
| Mean (SD) | 7.3 (17.8) | 11.6 (34.0) | 12.2 (35.4) | 7.9 (19.9) | 7.3 (18.4) | 12.6 (17.4) | 15.3 (36.7) |
| Median (IQR) | 3.6 (1.9, 6.7) | 4.7 (2.4, 9.9) | 4.8 (2.6, 9.8) | 2.9 (1.8, 6.7) | 2.7 (1.7, 4.7) | 6.7 (3.0, 14.7) | 6.6 (3.2, 13.2) |

[a] at any time during their admission episode

[b] use of special care unit codes 10 (medical intensive care nursing unit), 30 (combined medical/surgical intensive care nursing unit), 70 (pediatric intensive care nursing unit), or 80 (respiratory intensive care nursing unit)

[c] adjusted odds ratio (aOR) with 95% confidence interval (CI) adjusted for age (continuous, categorical, and their interaction), and sex. For respiratory syncytial virus, human metapneumovirus, and rhinovirus/enterovirus, models were also adjusted for calendar month, rurality, material deprivation, residential instability, dependency, and ethnic diversity neighbourhood quintiles.

[d] any evidence of respiratory resuscitation (1GZ30), respiratory ventilation (1GZ31), oxygenation of the respiratory system (1GZ32), local pharmacotherapy of the respiratory system (1GZ35), or management of some external appliance of the respiratory system (e.g. hyperbaric oxygen system, positive pressure ventilator, or positive pressure and expiratory pressure ventilator) (1GZ38) in any position

[e] calculated for the entire admission episode

SD–standard deviation; IQR–interquartile range

rhinovirus/enterovirus infections may have escaped recognition more easily than viruses like influenza, leading to less behavioural modification that would reduce transmission. School closures and re-openings would also have influenced transmission patterns of these viruses, particularly if outbreaks went unrecognized due to a disproportional focus on SARS-CoV-2. The resurgence of rhinovirus/enterovirus observed in other regions (e.g. California, South Korea) was also observed in Ontario for hospitalizations [12].

We observed a shift towards younger infants being admitted to hospital for RSV. These findings align with data from France and Iceland, the only two countries from a multinational study examining surveillance data from 17 countries during the first year after the start of the pandemic [11]. Due to the missed RSV season during the first year of the COVID-19

pandemic, reduced exposure may have resulted in fewer maternal antibodies being delivered *in-utero*, resulting in more hospitalizations occurring at a younger age among the 0-6-month old age group. Similarly, due to absent immunity from the missed season of RSV, older children within the age range of 2–5 years were more likely to be hospitalized during the resurgence than pre-pandemic seasons. Admissions with RSV or rhinovirus/enterovirus were less likely to comprise of patients age 65+ and more likely to comprise of young children (ages 2–5), which may be to continued adherence to various public health measures such as masking among the elderly amidst easing restrictions [28].

In many international studies, the first expected RSV season following the onset of the non-pharmacologic interventions used to combat the COVID-19 pandemic was not observed, but was followed by an off-season resurgence [29–32]. The combination of easing safeguards against SARS-CoV-2 infection back to pre-pandemic levels plus a lack of built-up immunity from prior exposure to these common viruses is expected to result in subsequent surges of infections occurring out-of-season and with increased morbidity [32–34]. School re-opening and lifted stay-at-home orders were found to be the most significant drivers for a rebound of RSV [31, 32]. In Ontario, public schools were opened for in-person learning at the start of the 2020/2021 school year (September 2020), and was only reverted back to teacher-led remote learning by January 2021 as SARS-CoV-2 infections surged yet again [35]. However, there was no sign of increased hospital activity due to RSV before remote learning began (Fig 3).

Reductions were also observed for influenza, but evidence of a resurgence was sparse at the time of writing [36–39]. In Australia, where the seasonal surge in admissions is typically observed in the summer months, surveillance reporting from sentinel hospital sites revealed a return to pre-pandemic levels of hospital admissions for the 2022/23 season (the third since the pandemic began) with a peak in spring [40]. It is therefore likely that the third influenza season in northern climates like Ontario will exhibit a similar pattern, arriving earlier than is typical as has been observed with RSV.

Throughout the COVID-19 pandemic, due to nurse and physician burn-out and absenteeism due to illness, hospitals are finding themselves understaffed and in some cases, have been forced to temporarily close EDs. At the time of writing, there is an anticipated surge of SARS-CoV-2 infected cases this coming fall/winter as in-person education resumes for the 2022/23 year. The cumulative effect of SARS-CoV-2, influenza, and RSV surges may overwhelm

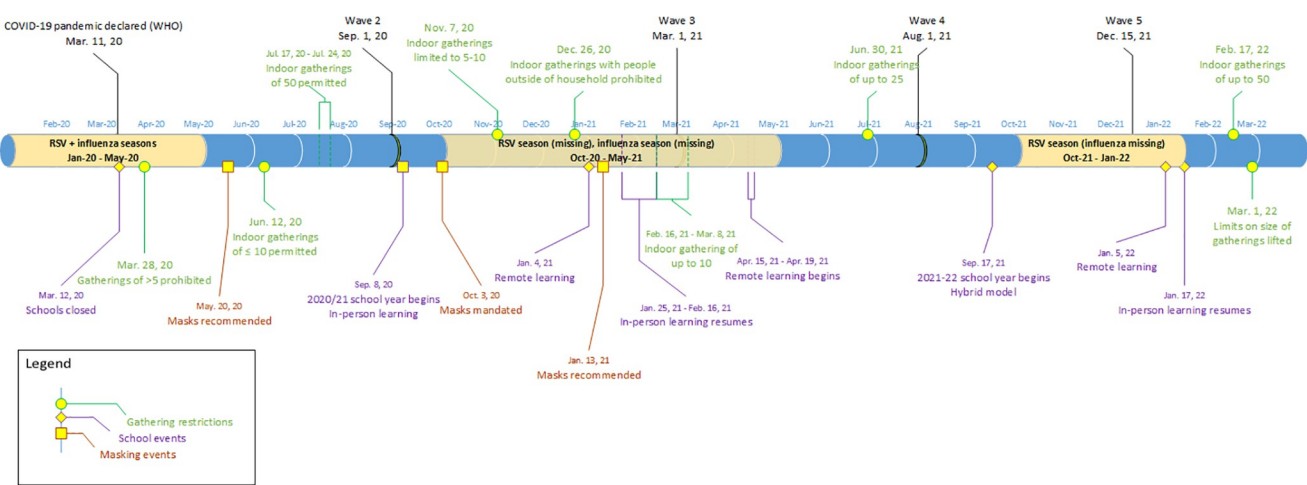

**Fig 3. Timeline of interventions.** Timing of restrictions and mandates related to gatherings, school events, and masking relative to the respiratory syncytial virus (RSV) seasons.

already taxed EDs and healthcare staff. Furthermore, the coexistence of SARS-CoV-2 and influenza in hospitalized patients was rare in the early pandemic, but co-infection is more likely in the upcoming 2022/23 influenza season and the implications of this on viral pathogenesis and patient outcomes remains unclear [38, 41, 42]. Alleviating strain on the healthcare system may include public health campaigns to promote influenza vaccination, further uptake of anti-RSV vaccination (maternal or among higher-risk infants and children) [43], and exploring RSV vaccination for older adults [44].

We did not observe a resurgence of hMPV, in contrast to reports from Western Australia, Israel, and the Netherlands [14, 45, 46]. Our study also found a return of hPINV earlier than expected, and a return to pre-pandemic admissions with common cold coronavirus and adenovirus-associated, although the numbers of events were small. Similar increases were observed in England [47].

### Limitations

There are several limitations of this study that should be considered. First, hospital administrative databases are not perfect. While ICD-10 coding has moderate sensitivity (73% for influenza and 69% for RSV), provincial systemic laboratory reporting to confirm respiratory virus diagnoses (e.g. OLIS) was unavailable for this dataset [18]. Second, testing practices for the broader array of respiratory viruses is not standardized in Ontario. The Public Health Ontario Laboratory will accept samples for broader respiratory multiplex testing from 1) all symptomatic hospitalized patients, 2) the first four symptomatic patients in an outbreak, 3) symptomatic patients tested in institutional settings; and 4) children <18 years old who present to the ED with compatible symptoms [48]. Hospitals have different arrays of test eligibility and breadth of respiratory viral testing based on local priorities. Third, because of the testing burden on laboratories during the COVID-19 pandemic, hospitals experiencing a higher burden of SARS-CoV-2 infections may have reduced their testing for other respiratory viruses based on local medical assessment, capacity of labs, financials, and competing demands [13]. Similar reductions in reporting have been described elsewhere [49]. We therefore expect the hospitalization rates attributable to non-SARS-CoV-2 respiratory viruses to be underestimated, particularly for ED visits since patients may be discharged without a test. Fourth, although this analysis is population-based from Ontario, the generalizability of these findings may be limited to jurisdictions with similar climates (e.g. due to alignment in the physical and social conditions conducive to transmission) and practice similar non-pharmacologic interventions. Anecdotally, Ontario took a relatively conservative stance on pandemic public health measures with a range of lockdowns, restrictions, and recommendations [35].

### Conclusion

In Ontario, Canada, there was a reduction in hospitalizations for RSV, influenza, hMPV, rhinovirus/enterovirus, hPINV, and adenoviruses at the onset of the COVID-19 pandemic. Hospital visits due to influenza remains absent, but we observed an off-peak resurgence of RSV presenting among younger infants and older children.

### Supporting information

**S1 Fig. Age histogram by virus and hospital visit type.** Histogram of age at admission or emergency department (ED) visit over the entire study period by virus diagnosis.
(DOCX)

**S2 Fig. Human parainfluenza virus.** Number of hospital admissions (A) and emergency department (ED) visits associated with human parainfluenza virus (hPINV).
(PDF)

**S3 Fig. Adenovirus.** Number of hospital admissions (A) and emergency department (ED) visits associated with adenovirus.
(PDF)

**S4 Fig. Common cold coronavirus.** Number of hospital admissions (A) and emergency department (ED) visits associated with the common cold coronavirus.
(PDF)

**S1 Table. Patient demographics by virus and hospital visit type.** Age and sex at the time of admission or emergency department visit associated with each virus.
(DOCX)

**S2 Table. Characteristics of emergency department visits with respiratory syncytial virus.** Factors associated with emergency department visit associated with respiratory syncytial virus during the COVID-19 era compared with the pre-COVID-19 era.
(DOCX)

**S3 Table. Characteristics of admissions with rhinovirus/enterovirus.** Factors associated with hospitalizations associated with rhinovirus/enterovirus during the COVID-19 era compared with the pre-COVID-19 era.
(DOCX)

**S4 Table. Main diagnostic code (most responsible diagnosis) associated with admission episode, by viral infection.** Number and percent of admission episodes by ICD-10 code, by virus type.
(DOCX)

## Acknowledgments

Parts of this material are based on data and information compiled and provided by the Canadian Institute of Health Information (CIHI). However, the analyses, conclusions, opinions and statements expressed herein are those of the author, and not necessarily those of CIHI.

We acknowledge support of the Ministry of Health and Long-Term Care in this report. All views expressed are those of the authors of this report and do not necessarily reflect those of Ontario or the Ministry.

## Author Contributions

**Conceptualization:** Steven Habbous, Susy Hota, Vanessa G. Allen, Erik Hellsten.

**Data curation:** Steven Habbous.

**Formal analysis:** Steven Habbous.

**Investigation:** Steven Habbous, Susy Hota, Vanessa G. Allen, Michele Henry, Erik Hellsten.

**Methodology:** Steven Habbous, Vanessa G. Allen, Erik Hellsten.

**Project administration:** Steven Habbous.

**Supervision:** Erik Hellsten.

**Visualization:** Steven Habbous.

**Writing – original draft:** Steven Habbous.

**Writing – review & editing:** Steven Habbous, Susy Hota, Vanessa G. Allen, Michele Henry, Erik Hellsten.

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
