## [Decision Letter · Decision Letter 0]

2 Mar 2023

PONE-D-22-35031Hospitalizations and emergency department visits for respiratory viruses in the time of the COVID-19 pandemic in Ontario, CanadaPLOS ONE

Dear Dr. Habbous,

Thank you for submitting your manuscript to PLOS ONE. We apologize for the long delay and appreciate your patience as it took some time to ascertain reviews. After careful consideration, we feel that it has merit but does not fully meet PLOS ONE’s publication criteria as it currently stands. Therefore, we invite you to submit a revised version of the manuscript that addresses the points raised during the review process.

We look forward to receiving your revised manuscript.

Kind regards,

Andrew G Wu

Academic Editor

PLOS ONE

Journal Requirements:

Reviewers' comments:

Reviewer's Responses to Questions

**Comments to the Author**

1. Is the manuscript technically sound, and do the data support the conclusions?

Reviewer #1: Partly

Reviewer #2: Yes

2. Has the statistical analysis been performed appropriately and rigorously? 

Reviewer #1: Yes

Reviewer #2: I Don't Know

3. Have the authors made all data underlying the findings in their manuscript fully available?

Reviewer #1: No

Reviewer #2: Yes

4. Is the manuscript presented in an intelligible fashion and written in standard English?

Reviewer #1: Yes

Reviewer #2: Yes

5. Review Comments to the Author

Reviewer #1: Thank you for the opportunity to review this manuscript. Habbous et al use administrative code data for a number of common community respiratory viruses in Ontario Canada and describe the change in trends before and during the COVID-19 pandemic. They find that there are important differences by viral species that reflect similar findings reported in other locations. The topic of COVID-19 pandemic effects on common respiratory viruses has been well documented in different locations but with slight differences that are likely affected by differences in public heatlh interventions, community attitudes and behaviors towards COVID-19 as well as climate. As such, additional publications such as this I believe still has relevance and would be of interest to the readers of PLOSOne. However, as the authors highlight, the biggest limitation with this study is the use of administrative codes. Rather than dive further into the limitations of this, I have some suggestions below. Additionally, I have provide comments, questions and suggestions that I hope will be helpful to the authors as they consider revision.

Major Consideration

1. As mentioned, the biggest limitation to this study is the use of administrative codes. Testing surveillance already has limitations caused by pandemic related health seeking behaviors and supply chain issues. Administrative codes would have even more bias given that it relies on the coder to accurately depict the type of respiratory viral infection. I would strongly suggest that the authors be as upfront about this limitation as possible and put it in the title. I suggest “Changes in Hospitalization and Emergency Department Respiratory Viral Diagnosis Trends Before and During the COVID-19 Pandemic, Ontario, Canada.”

2. In the introduction, the authors highlight a number of reasons why there may have been changes in respiratory viral circulation during the pandemic but failed to mention the possibility of viral interference. I suggest including this and also adding a citation. I also suggest that testing behaviors due to heightened awareness of respiratory viruses or in some cases prioritizing COVID-19 may have influenced diagnoses of other respiratory viruses. Reagent limitations and supply chain issues also played a role in these.

3. A number of studies on this topic have been published that have provided a comprehensive overview of the literature out there. I would suggest also including this study in the introduction: https://www.nature.com/articles/s41579-022-00807-9 and it can also be added to the discussion as this paper includes a discussion on hospitalization and ED visits.

4. On page 7 when the authors describe the changes in age-groups among children, I suggest also pointing out that the proportion of adults > 65 + remained suppressed in contrast to the increase in pediatric cases during the pandemic. It would be worth then to elaborate on this in the discussion. Falsey et al attributed this change to the return of children to school and the ongoing mask use in those 65 years and older given the higher risk for sever outcomes related to COVID-19: https://academic.oup.com/jid/article/227/1/83/6783142

5. In the discussion, the authors provide highlight that rhino/enteroviruses may transmit more efficiently by fomites and surface contamination as the reason for their increases later in the pandemic. However, other reasons exist as described in the discussion here: https://academic.oup.com/jid/article/226/Supplement_3/S304/6617600?searchresult=1

6. It is not clear to me how the authors established the predicted levels of hospital admissions and emergency department visits in Figure 1 and 2. This should be described in the methods.

7. Other viruses were highlighted in the results other than influenza, RSV, rhino/enterovirus. Suggest putting the study results in context to what is published on others such as HPIV, HMV, Adenovirus.

Other considerations

1. Rather than referring to endemic coronaviruses as “seasonal coronavirus” I suggest using the term common cold coronaviruses: https://pubmed.ncbi.nlm.nih.gov/36052654/

2. Page 7 there is a typo when the authors compare the percentages of hospitalized children before and during the pandemic (should be 8% instead of 32%).

3. Figure 1 – subtitles should be hospital admissions and not emergency admissions.

4. Table 2 – age groups should not have overlapping ages (e.g. 0-6 months, 6-24 months)

5. eFigure S1 should indicate whether these data are before, during the pandemic or over the study period.

Reviewer #2: To improve the quality of the manuscript, I would suggest the authors:

-to include the main reason and relative % of hospitalization

-to consider some variables (such as intensive care unit occupation or oxygen supplementation): they may influence hospitalization lenght and contribute to a high hospitalization costs

- to include in the discussion a comparison with other reports ( for example: doi: 10.3390/ijerph192315455) supporting the need of protection in infants and eldery ( doi: 10.1186/s12982-021-00104-5 doi: 10.1056/NEJMoa2116154)

6. PLOS authors have the option to publish the peer review history of their article (what does this mean?). If published, this will include your full peer review and any attached files.

Reviewer #1: No

Reviewer #2: **Yes: **Elena Bozzola

---

## [Author Response · Author response to Decision Letter 0]

8 May 2023

March 20, 2023

Manuscript PONE-D-22-35031 

Title: “Hospitalizations and emergency department visits for respiratory viruses in the time of the COVID-19 pandemic in Ontario, Canada”

Running head: Respiratory viruses during the COVID-19 pandemic

Dear Dr. Wu,

We thank you, the editorial team, and the reviewers for taking the time to review this manuscript. We provide point-by-point responses to the comments provided below and have tracked changes in the manuscript. All other edits were grammatical in nature to reduce the word count

Sincerely,

Steven Habbous, PhD 

Epidemiologist 

Ontario Health (Strategic Analytics) 

Email: steven.habbous@ontariohealth.ca

 

Hospitalizations and emergency department visits for respiratory viruses in the time of the COVID-19 pandemic in Ontario, Canada

PLOS ONE

Dear Dr. Habbous,

Thank you for submitting your manuscript to PLOS ONE. We apologize for the long delay and appreciate your patience as it took some time to ascertain reviews. After careful consideration, we feel that it has merit but does not fully meet PLOS ONE’s publication criteria as it currently stands. Therefore, we invite you to submit a revised version of the manuscript that addresses the points raised during the review process.

We look forward to receiving your revised manuscript.

Kind regards,

Andrew G Wu

Academic Editor

PLOS ONE

Journal Requirements:

Response: We have reviewed the requirements and made appropriate edits.

Response: We have added the data availability statement to the cover letter. Due to legal and privacy issues, we are prohibited from making the data publicly available.

Response: This is my first time submitting to a PLoS journal and have no funding for this work. However, my library contacts at Western University assure me that there should be no issue given the funding agreement between PLoS and Western University.

Reviewers' comments:

Reviewer's Responses to Questions

Comments to the Author

1. Is the manuscript technically sound, and do the data support the conclusions?

Reviewer #1: Partly

Reviewer #2: Yes

2. Has the statistical analysis been performed appropriately and rigorously? 

Reviewer #1: Yes

Reviewer #2: I Don't Know

3. Have the authors made all data underlying the findings in their manuscript fully available?

Reviewer #1: No

Reviewer #2: Yes

4. Is the manuscript presented in an intelligible fashion and written in standard English?

Reviewer #1: Yes

Reviewer #2: Yes

5. Review Comments to the Author

Reviewer 1 comments

Reviewer #1: 

Thank you for the opportunity to review this manuscript. Habbous et al use administrative code data for a number of common community respiratory viruses in Ontario Canada and describe the change in trends before and during the COVID-19 pandemic. They find that there are important differences by viral species that reflect similar findings reported in other locations. The topic of COVID-19 pandemic effects on common respiratory viruses has been well documented in different locations but with slight differences that are likely affected by differences in public health interventions, community attitudes and behaviors towards COVID-19 as well as climate. As such, additional publications such as this I believe still has relevance and would be of interest to the readers of PLOSOne. However, as the authors highlight, the biggest limitation with this study is the use of administrative codes. Rather than dive further into the limitations of this, I have some suggestions below. Additionally, I have provide comments, questions and suggestions that I hope will be helpful to the authors as they consider revision.

Major Consideration

Comment #1: As mentioned, the biggest limitation to this study is the use of administrative codes. Testing surveillance already has limitations caused by pandemic related health seeking behaviors and supply chain issues. Administrative codes would have even more bias given that it relies on the coder to accurately depict the type of respiratory viral infection. I would strongly suggest that the authors be as upfront about this limitation as possible and put it in the title. I suggest “Changes in Hospitalization and Emergency Department Respiratory Viral Diagnosis Trends Before and During the COVID-19 Pandemic, Ontario, Canada.”

Response: Thank you for the suggestion, we have changed the title accordingly. We agree that the wording of the new title is more aligned with the paper’s methods.

Comment #2: In the introduction, the authors highlight a number of reasons why there may have been changes in respiratory viral circulation during the pandemic but failed to mention the possibility of viral interference. I suggest including this and also adding a citation. I also suggest that testing behaviors due to heightened awareness of respiratory viruses or in some cases prioritizing COVID-19 may have influenced diagnoses of other respiratory viruses. Reagent limitations and supply chain issues also played a role in these.

Response: These are all excellent points and we have listed them as possible reasons for reduced hospital activity in the introduction. This part of the introduction now reads as follows:

“Since the start of the COVID-19 pandemic, the number of people affected by influenza, RSV, and the broader array of respiratory viruses decreased in many countries.4–9 Although this may be partially driven by viral interference caused by COVID-19 infection, reduced testing due to reagent shortages, and laboratory burden, the main drivers are likely modifications in human behavior as a result of public health measures (e.g. restrictions, lockdowns) and individual risk-reduction tactics (e.g. mask-wearing, avoiding crowded spaces, increased hand hygiene) to combat the COVID-19 pandemic.10,11”

We have also added to the discussion the following:

“It is also possible that viral interference (e.g. infection with COVID-19 temporarily guarded against other infections) may have prevented some hospitalizations related to influenza or RSV, although perhaps less with rhinovirus.”

Comment #3: A number of studies on this topic have been published that have provided a comprehensive overview of the literature out there. I would suggest also including this study in the introduction: https://www.nature.com/articles/s41579-022-00807-9 and it can also be added to the discussion as this paper includes a discussion on hospitalization and ED visits.

Response: This is a very thorough review. We have added this citation to the introduction upon listing the aforementioned reasons for changes in respiratory viral circulation. We also cite this review in the discussion upon expounding on rhinovirus/enterovirus.

Comment #4: On page 7 when the authors describe the changes in age-groups among children, I suggest also pointing out that the proportion of adults > 65 + remained suppressed in contrast to the increase in pediatric cases during the pandemic. It would be worth then to elaborate on this in the discussion. Falsey et al attributed this change to the return of children to school and the ongoing mask use in those 65 years and older given the higher risk for severe outcomes related to COVID-19: https://academic.oup.com/jid/article/227/1/83/6783142

Response: We have added the following to the discussion, referencing this study: 

“Admissions with RSV or rhinovirus/enterovirus were less likely to comprise of patients age 65+ and more likely to comprise of young children (ages 2-5), which may be to continued adherence to various public health measures such as masking among the elderly amidst easing restrictions (Falsey et al).”

Comment #5: In the discussion, the authors provide highlight that rhino/enteroviruses may transmit more efficiently by fomites and surface contamination as the reason for their increases later in the pandemic. However, other reasons exist as described in the discussion here: https://academic.oup.com/jid/article/226/Supplement_3/S304/6617600?searchresult=1

Response: The discussion from this article was very helpful. We have added the following to the discussion, adding appropriate references throughout:

“It is possible that viruses such as rhinoviruses and enteroviruses transmit more efficiently than other respiratory viruses, explaining their rising impacts later in the pandemic. Rhinoviruses are non-enveloped viruses, prolonging their survival on inanimate surfaces and enabling transmission by fomites and surface contact (PMC4462923, 30373527). Masking is also less effective against rhinoviruses than influenza (PMC8238571).”

Comment #6: It is not clear to me how the authors established the predicted levels of hospital admissions and emergency department visits in Figure 1 and 2. This should be described in the methods.

Response: Thank you for noticing this. We have added the following to the “Statistics and privacy” subsection of the methods:

“The predicted levels of admissions or emergency department visits were constructed using a linear regression on the weekly number of visits between January 2017 and February 2020 using calendar year, month, and holiday weeks as predictors. This was then extrapolated into the COVID-19 era.”

Comment 7: Other viruses were highlighted in the results other than influenza, RSV, rhino/enterovirus. Suggest putting the study results in context to what is published on others such as HPIV, HMV, Adenovirus.

Response: Great suggestion. We have added the following to the end of the discussion before the limitations:

“We did not observe a resurgence of hMPV, in contrast to reports from Western Australia, Israel, and the Netherlands.(PMC9612024, 35472530, PMC8847107) Our study also found a return of hPINV earlier than expected, and a return to pre-pandemic admissions with seasonal coronavirus and adenovirus-associated, although the numbers of events were small. Similar increases were observed in England.( PMC8591975)”

Other considerations

Comment 8: Rather than referring to endemic coronaviruses as “seasonal coronavirus” I suggest using the term common cold coronaviruses: https://pubmed.ncbi.nlm.nih.gov/36052654/

Response: We have changed the language throughout (and cited this paper).

Comment 9: Page 7 there is a typo when the authors compare the percentages of hospitalized children before and during the pandemic (should be 8% instead of 32%).

Response: Good catch, we have made this correction.

Comment 10: Figure 1 – subtitles should be hospital admissions and not emergency admissions.

Response: Thank you for the suggestion. We have removed the subtitles from the figures, as these have become redundant

Comment 11: Table 2 – age groups should not have overlapping ages (e.g. 0-6 months, 6-24 months)

Response: We have specified the cut-points for Table 2 and the supplementary tables.

Comment 12: eFigure S1 should indicate whether these data are before, during the pandemic or over the study period.

Response: These supplementary figures were over the entire study period. We have added a caption to clarify this.

Reviewer #2 comments

To improve the quality of the manuscript, I would suggest the authors:

Comment 1: to include the main reason and relative % of hospitalization

Response: Thank you for the suggestion. We have added a supplementary table to indicate, for each virus, the most responsible diagnosis associated with that admission (eTable S3). 

Comment2: to consider some variables (such as intensive care unit occupation or oxygen supplementation): they may influence hospitalization length and contribute to a high hospitalization costs

Response: This is a great suggestion. We have added total admission length-of-stay, use of ICU, and provision of respiratory ventilation or oxygenation as outcomes, by viral type (new Table 3). We have added the following to the new “outcomes” subsection of the methods:

“During the admission episode, we examined the utilization of intensive care units (ICU) (special care unit codes 10, 30, 70, or 80) and use of respiratory support (1GZ30, 1GZ31, 1GZ32, 1GZ35, 1GZ38). Total length of stay was computed, in days, at the level of the entire admission episode.”

We have also added the following to the new “outcomes” subsection of the results: 

Use of ICU during the admission episode was highest for hPINV (23%), followed by rhinovirus/enterovirus (18%), hMPV (14.6%), influenza (11.3%), adenovirus (9.7%), and RSV (9.2%) (Table 3). The high rate of ICU use also observed among common cold coronaviruses (18%) may be driven by misclassification with the novel COVID-19 (eFigure S4; eTable S3). Respiratory support during the admission episode was highest for admissions with common-cold coronavirus (17%) and hPINV (16%), followed by hMPV (10%), rhinovirus/enterovirus (8.9%), influenza virus (7.5%), and RSV (5.8%). Total length-of-stay was highest for hPINV (median 6.6 days), hMPV (median 4.8 days), and influenza virus (median 4.7 days).

ICU use in the COVID-19 era was more likely for admissions with hMPV [OR 1.89 (1.13-3.16)]. Use of respiratory support was trending higher in the COVID era for hMPV [OR 1.64 (0.91-2.94)], but decreased for RSV [OR 0.61 (0.47-0.78)] and rhinovirus/enterovirus [OR 0.53 (0.34-0.82)] (Table 3).

Comment3: to include in the discussion a comparison with other reports ( for example: doi: 10.3390/ijerph192315455) supporting the need of protection in infants and elderly ( doi: 10.1186/s12982-021-00104-5 doi: 10.1056/NEJMoa2116154)

Response: Thank you for these references, we have added these references to the discussion: “Alleviating strain on the healthcare system may include public health campaigns to promote influenza vaccination, further uptake of anti-RSV vaccination (maternal or among higher-risk infants and children)43, explore RSV vaccination for older adults.44”

6. PLOS authors have the option to publish the peer review history of their article (what does this mean?). If published, this will include your full peer review and any attached files.

Do you want your identity to be public for this peer review? For information about this choice, including consent withdrawal, please see our Privacy Policy.

Reviewer #1: No

Reviewer #2: Yes: Elena Bozzola

---

## [Decision Letter · Decision Letter 1]

31 May 2023

PONE-D-22-35031R1Changes in hospitalizations and emergency department respiratory viral diagnosis trends before and during the COVID-19 pandemic in Ontario, CanadaPLOS ONE

Dear Dr. Habbous,

Thank you for submitting your manuscript to PLOS ONE. After careful consideration, we feel that it has merit but does not fully meet PLOS ONE’s publication criteria as it currently stands. Therefore, we invite you to submit a revised version of the manuscript that addresses the points raised during the review process. I'm also pleased to report that I am happy to accept your manuscript for publication once the minor revisions have been addressed.

We look forward to receiving your revised manuscript.

Kind regards,

Andrew G Wu

Academic Editor

PLOS ONE

Journal Requirements:

Reviewers' comments:

Reviewer's Responses to Questions

**Comments to the Author**

1. If the authors have adequately addressed your comments raised in a previous round of review and you feel that this manuscript is now acceptable for publication, you may indicate that here to bypass the “Comments to the Author” section, enter your conflict of interest statement in the “Confidential to Editor” section, and submit your "Accept" recommendation.

Reviewer #1: All comments have been addressed

Reviewer #2: All comments have been addressed

2. Is the manuscript technically sound, and do the data support the conclusions?

Reviewer #1: Yes

Reviewer #2: Yes

3. Has the statistical analysis been performed appropriately and rigorously? 

Reviewer #1: Yes

Reviewer #2: I Don't Know

4. Have the authors made all data underlying the findings in their manuscript fully available?

Reviewer #1: No

Reviewer #2: Yes

5. Is the manuscript presented in an intelligible fashion and written in standard English?

Reviewer #1: Yes

Reviewer #2: Yes

6. Review Comments to the Author

Reviewer #1: Thank you for taking the time to consider my comments and suggestions. I appreciate all that was done to address my concerns. A couple of minor points:

1. In the introduction, detection of of respiratory viruses during the pandemic may have also been affected by testing behavior. Early on, people may have feared going to the clinic for testing or later, people no longer cared to know what their diagnosis was unless they were sick enough to be hospitalized.

2. All instances in which you're referring to the infection, it's probably more accurate to say SARS-CoV-2 infection. When referring to the disease you can use COVID-19. Suggest not using COVID-19 infection.

3. Table 2 - The age group overlap is still a little confusing to me. I am not familiar with the use of parentheses and brackets to represent inclusivity or exclusivity. I suspect other readers may also not be familiar with this. Please adjust to be clear or be clear in the footnotes about how these are used.

4. If you are short on word count, you could consider abbreviating common cold coronavirus as ccCoV

Reviewer #2: Authors provided an answer to my concern. The manuscript is suitable for pubblication in the present form

7. PLOS authors have the option to publish the peer review history of their article (what does this mean?). If published, this will include your full peer review and any attached files.

Reviewer #1: No

Reviewer #2: No

---

## [Author Response · Author response to Decision Letter 1]

31 May 2023

Reviewer 1 comments

Thank you for taking the time to consider my comments and suggestions. I appreciate all that was done to address my concerns. A couple of minor points:

Comment #1: In the introduction, detection of respiratory viruses during the pandemic may have also been affected by testing behavior. Early on, people may have feared going to the clinic for testing or later, people no longer cared to know what their diagnosis was unless they were sick enough to be hospitalized.

Response: Thank you for the comment, we have added this possibility to the introduction along with a relevant reference (#5; Moore et al).

Comment #2: All instances in which you're referring to the infection, it's probably more accurate to say SARS-CoV-2 infection. When referring to the disease you can use COVID-19. Suggest not using COVID-19 infection.

Response: Great point. We went through the manuscript to ensure whenever we’re referring to the infection, we are using SARS-CoV-2 instead of COVID-19. When speaking more generally (e.g. the COVID-19 era; the COVID-19 pandemic), we retained the use of COVID-19.

Comment #3: Table 2 - The age group overlap is still a little confusing to me. I am not familiar with the use of parentheses and brackets to represent inclusivity or exclusivity. I suspect other readers may also not be familiar with this. Please adjust to be clear or be clear in the footnotes about how these are used.

Response: Thank you for the suggestion, we have clarified this in the tables and the supplement.

Comment #4: If you are short on word count, you could consider abbreviating common cold coronavirus as ccCoV

Response: Thank you for the suggestion, but I believe the word count is okay.

Reviewer 2 comments

Comment: Authors provided an answer to my concern. The manuscript is suitable for publication in the present form

Response: Thank you.

---

## [Editor Report · Decision Letter 2]

5 Jun 2023

Changes in hospitalizations and emergency department respiratory viral diagnosis trends before and during the COVID-19 pandemic in Ontario, Canada

PONE-D-22-35031R2

Dear Dr. Habbous,

We’re pleased to inform you that your manuscript has been judged scientifically suitable for publication and will be formally accepted for publication once it meets all outstanding technical requirements.

Kind regards,

Andrew G Wu

Academic Editor

PLOS ONE

---

## [Editor Report · Acceptance letter]

8 Jun 2023

PONE-D-22-35031R2 

Changes in hospitalizations and emergency department respiratory viral diagnosis trends before and during the COVID-19 pandemic in Ontario, Canada 

Dear Dr. Habbous:

I'm pleased to inform you that your manuscript has been deemed suitable for publication in PLOS ONE. Congratulations! Your manuscript is now with our production department. 

Kind regards, 

on behalf of

Dr. Andrew G Wu 

Academic Editor

PLOS ONE